# Dual versus single vessel normothermic *ex vivo* perfusion of rat liver grafts using metamizole for vasodilatation

**Felix Claussen**[1], **Joseph M. G. V. Gassner**[1], **Simon Moosburner**[1], **David Wyrwal**[1], **Maximilian Nösser**[1], **Peter Tang**[1], **Lara Wegener**[1], **Julian Pohl**[1], **Anja Reutzel-Selke**[1], **Ruza Arsenic**[2], **Johann Pratschke**[1], **Igor M. Sauer**[1]*, **Nathanael Raschzok**[1,3]

**1** Department of Surgery, Campus Charité Mitte | Campus Virchow Klinikum, Charité–Universitätsmedizin Berlin, Corporate Member of Freie Universität Berlin, Humboldt- Universität zu Berlin and Berlin Institute of Health, Berlin, Germany, **2** Institute of Pathology, Charité–Universitätsmedizin Berlin, Corporate Member of Freie Universität Berlin, Humboldt- Universität zu Berlin and Berlin Institute of Health, Berlin, Germany, **3** BIH Charité Clinician Scientist Program, Berlin Institute of Health (BIH), Berlin, Germany

* igor.sauer@charite.de

**Data Availability Statement:** All relevant data are within the paper and its Supporting Information file.

## Abstract

### Background

Normothermic *ex vivo* liver perfusion (NEVLP) is a promising strategy to increase the donor pool in liver transplantation. Small animal models are essential to further investigate questions regarding organ preservation and reconditioning by NEVLP. A dual vessel small animal NEVLP (dNEVLP) model was developed using metamizole as a vasodilator and compared to conventional portovenous single vessel NEVLP (sNEVLP).

### Methods

Livers of male Wistar rats were perfused with erythrocyte-supplemented culture medium for six hours by either dNEVLP via hepatic artery and portal vein or portovenous sNEVLP. dNEVLP was performed either with or without metamizole treatment. Perfusion pressure and flow rates were constantly monitored. Transaminase levels were determined in the perfusate at the start and after three and six hours of perfusion. Bile secretion was monitored and bile LDH and GGT levels were measured hourly. Histopathological analysis was performed using liver and bile duct tissue samples after perfusion.

### Results

Hepatic artery pressure was significantly lower in dNEVLP with metamizole administration. Compared to sNEVLP, dNEVLP with metamizole treatment showed higher bile production, lower levels of transaminases during and after perfusion as well as significantly lower necrosis in liver and bile duct tissue. Biochemical markers of bile duct injury showed the same trend.

**Funding:** This work was funded by institutional financial support of the Charité – Universitätsmedizin Berlin, the German Research Foundation (grant number: RA 3044/3-1) and by a Kickbox Seed Grant of the Einstein Center for Regenerative Therapies. Nathanael Raschzok is fellow of the BIH Charité Clinician Scientist Program funded by the Charité – Universitätsmedizin Berlin and the Berlin Institute of Health.

**Competing interests:** The authors have declared that no competing interests exist.

**Abbreviations:** ALT, alanine aminotransferase; AST, aspartate aminotransferase; dNEVLP, dual vessel normothermic *ex vivo* liver perfusion; ECD, extended criteria donor; GGT, gamma glutamyltransferase; H&E, haematoxylin and eosin; HA, hepatic artery; HTK, histidine-tryptophan-ketoglutarate solution; LDH, lactate dehydrogenase; NEVLP, normothermic *ex vivo* liver perfusion; PV, portal vein; sNEVLP, single vessel Normothermic *ex vivo* liver perfusion; TUNEL, terminal deoxynucleotidyl transferase dUTP nick end labeling.

## Conclusion

Our miniaturized dNEVLP system enables normothermic dual vessel rat liver perfusion. The administration of metamizole effectively ameliorates arterial vasospasm allowing for six hours of dNEVLP, with superior outcome compared to sNEVLP.

## Introduction

Liver transplantation is still the only curative treatment option for end stage liver disease. However, there has been an increasing mismatch of organ supply and demand in recent years [1]. The reasons for this development are manifold including progressive aging of the population and increased prevalence of non-alcoholic fatty liver disease in many countries [2, 3]. The severe organ shortage makes the use of marginal organs from so-called extended criteria donors (ECD) necessary [4]. ECDs are donors that do not meet the usual criteria for liver transplantation and are for example old or severely overweight [5, 6]. Data of the *United Network for Organ Sharing* show an increasing number of donors that are of high age or severely overweight [7]. Indeed, in Germany, up to 75% of liver grafts transplanted fulfil at least one *Eurotransplant* criterion for extended criteria donors [8]. However, it has been shown, that marginal organs from ECDs perform significantly worse and show higher rates of post-transplantation complications than organs from donors matching the usual transplantation criteria [9, 10]. This is commonly attributed to their increased susceptibility to ischemia-reperfusion injury after cold storage and consecutive warm reperfusion [10, 11].

The necessity of using ECD liver grafts, resulting in poorer patient outcome after cold static storage and transplantation, calls for new strategies for organ preservation. In recent years normothermic *ex vivo* liver perfusion (NEVLP) has been proven to be a useful alternative to static cold storage of liver grafts [12]. Several studies have demonstrated that NEVLP improves preservation of both fit and marginal liver grafts compared to static cold storage [13–15]. Besides its beneficial effects on the preservation of donor organs NEVLP, unlike subnormothermic or hypothermic *ex vivo* liver perfusion, also offers the opportunity to metabolically characterize the graft prior to transplantation and to perform pharmacological interventions that rely on a fully functional metabolism and near-to-physiological conditions. Such interventions could improve the quality of marginal liver grafts that are otherwise not acceptable for transplantation [12, 16, 17].

In order to develop and investigate organ recovery strategies based on NEVLP, animal models are needed. Since porcine models are costly and require highly developed infrastructures, small animal models seem more feasible for this task. Several models for *ex vivo* machine perfusion of rodent livers have been introduced in recent years. The majority of these models propose single vessel perfusion through the portal vein (PV). However, considering the physiological situation, a dual vessel approach, realizing perfusion through both the PV and the hepatic artery (HA), seems more coherent. Also, a dual vessel small animal NEVLP model would correspond to the clinical situation more adequately since NEVLP devices for human liver grafts also employ dual vessel perfusion [18–20].

We developed a dual vessel normothermic *ex vivo* liver perfusion (dNEVLP) model for rat livers that would sustain stable perfusion conditions for a perfusion period of six hours. We developed a flow-controlled perfusion system using metamizole to mitigate arterial vasospasm and control arterial perfusion pressure. Amongst other qualities, metamizole is known to have spasmolytic effects and has been shown to effectively ameliorate arterial vasospasm of the HA

of the rat [21, 22]. Subsequently we compared our previously established single vessel NEVLP model (sNEVLP) to our new dNEVLP model [23].

## Methods

### Perfusion setup

The laboratory-scaled sNEVLP setup, as initially described by Gassner et al. [23], consisted of a custom-made glass perfusion chamber with multiple inlets (Glass Gaßner GmbH, Munich, Germany). A flow-controlled roller pump provided a continuous flow through the portal vein. Flow was set to 1 mL/min/g liver weight through the portal vein. A pressure sensor ensured continuous monitoring of the portal venous pressure. Average initial pressure was 5.65 mmHg. Portovenous pressures up to 9 mmHg were considered physiological. Pressures were continuously recorded with BDAS 2.0 software (Harvard Apparatus, Holliston, MA, USA). Gas exchange was ensured by a silicon membrane oxygenator (Radnoti, Dublin, Ireland) with a priming volume of 10 mL and 90% oxygen atmosphere. A glass bubble trap prevented air embolization and worked as a *Windkessel* to ensure laminar flow through the portal vein. A dialysis circuit was diverted from the main perfusion circuit directly after the perfusion chamber. Another roller pump brought perfusate to the dialysis cartridge with a constant flow of 10 mL/min. 500 mL of Ci-Ca dialysate K2 (Fresesnius Kabi, Bad Homburg, Germany) substituted with 12 mM glycin were used. Dialysate flow of 10 mL/min was generated by two roller pumps, one up- and one downstream from the dialysis cartridge, allowing flow adjustments to counteract volume shifts. 1000 IE/h Heparin (Rotexmedica, Trittau, Germany) and 500 μL/h of 1.2 M glycin (45 mg/h) were continuously infused using a syringe driver (*Perfusor®*, *B. Braun Melsungen*, Melsungen, Germany, **Fig 1A**).

For dual vessel NEVLP (dNEVLP) a second flow-controlled roller pump, installed downstream to the first one, diverted a defined volume from the main circuit, generating a pulsatile flow through the hepatic artery. Flow was set to 1.1 mL/min/g liver weight for the first roller pump and 0.1 mL/min/g liver weight for the second roller pump, generating a flow of 1 mL/min/g liver weight through the portal vein and 0.1 mL/min/g liver weight through the hepatic artery. A second pressure sensor allowed continuous monitoring of the arterial pressure. Average initial arterial pressure was 48.8 mmHg and pressures up to 110 mmHg were considered physiological. The rest of the perfusion circuit was set up in the exact same manner as described for sNEVLP (**Fig 1B**).

In metamizole treatment groups, 100 mg boluses of metamizole sodium (*Winthrop Arzneimittel GmbH*, Frankfurt am Main, Germany) were administered into the hepatic artery either hourly or on demand at a pressure cutoff of 110 mmHg.

### Animals and group protocols

Male Wistar rats were purchased from *Janvier* (Le Genest-Saint-Isle, France). To allow for adequate acclimatisation, animals were kept on a 12-hour light-dark circle for a minimum of one week. All procedures were conducted within a weight range of 280–350 g and with the approval of the local authorities for animal welfare and testing (LaGeSo Berlin, O0365/11). Rats were randomly assigned to 4 groups: dNEVLP without metamizole treatment (dNEVLP⁻ᴹ, n = 4), dNEVLP with hourly administration of metamizole beginning one hour after perfusion start (dNEVLPᴹᴴ, n = 4), dNEVLP with administration of metamizole on demand at a pressure cut-off of 110 mmHg (dNEVLPᴹᴾ, n = 4) and sNEVLP (sNEVLP, n = 4) as baseline control. In a first step the three dNEVLP groups were compared. Subsequently the best dNEVLP group was compared to the sNEVLP group.

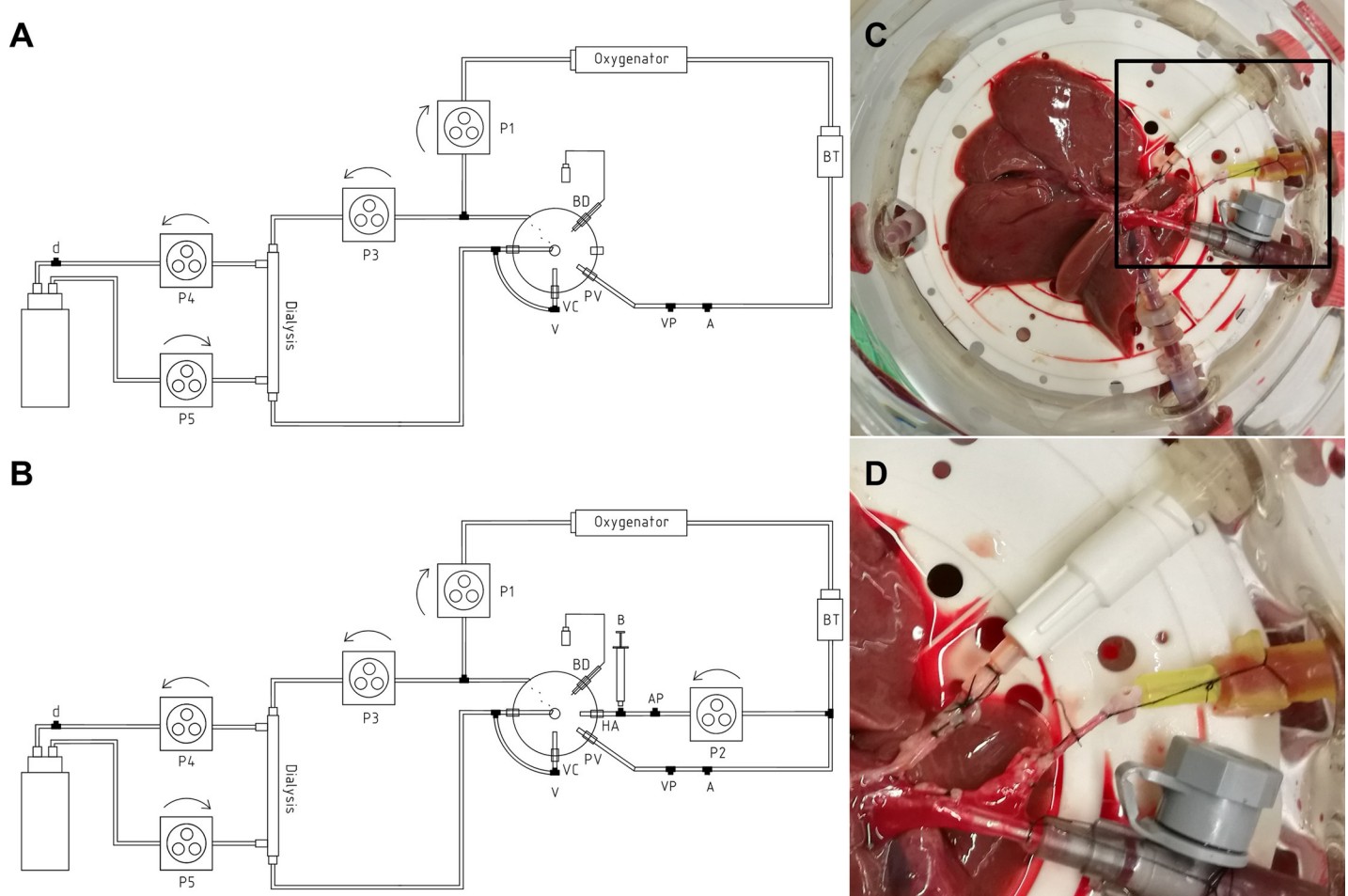

**Fig 1. Perfusion setups.** (A) Technical drawing of the sNEVLP setup consisting of the perfusion chamber with inlets for the portal vein (PV) and outlets for the vena cava (VC) and bile duct (BD), an oxygenator, a bubble trap (BT), a dialysis circuit and four roller pumps (P1, P3-5). Several outlets within the perfusion circuit allowed for sample collection of the arterial (A) and venous (V) perfusate and dialysate (d) and for measurement of the portovenous pressure (PV). (B) Technical drawing of the dNEVLP setup with an additional inflow for the hepatic artery (HA), a fifth roller pump (P2) and another two outlets for metamizole bolus application (B) and measurement of the arterial pressure (AP). (C) Liver in the perfusion chamber (dNEVLP) with cannulated bile duct, hepatic artery, portal vein and infrahepatic vena cava (top to bottom). (D) Close-up on cannulated hepatic artery (yellow cannula) witch patch from abdominal aorta.

## Surgical procedures

General anaesthesia was performed on animals using isoflurane inhalation and subcutaneous injections of 100 mg/kg metamizole (*Winthrop Arzneimittel GmbH*, Frankfurt am Main, Germany) and 12 mg/kg ketamine (*CP-Pharma*, Burgdorf, Germany). Subsequently, the abdominal cavity was opened. The liver was freed from its ligaments and the bile duct was catheterized using a custom-built catheter. The gastroduodenal, left gastric and splenic artery were ligated. 1 mL of Ringer solution supplemented with 500 IE Heparin (*Rotexmedica*, Trittau, Germany) was injected into the abdominal vena cava inferior. The abdominal aorta and the portal vein were cannulated for blood collection and later flushing. In the dNEVLP groups the hepatic artery was cannulated through an aortic patch (**Fig 1C and 1D**). The thoracic cavity was opened, and the thoracic aorta clamped. The liver was flushed via both, the aorta and the portal vein with 20 mL of 4°C cold HTK-solution (*Dr. Franz Köhler Chemie GmbH*, Bensheim, Germany), supplemented with 12mM glycine. Time between blood collection and cold flushing of the liver (warm ischemia time, WIT) did not exceed 15 minutes. The right

suprarenal vein, the oesophageal veins and the suprahepatic vena cava were ligated. A custom-made cannula was inserted into the infrahepatic vena cava. The liver was completely mobilised and then transferred into a pre-weighed container filled with cold HTK-solution supplemented with 12 mM glycine.

## Perfusion procedure

The HTK solution was flushed out of the liver via the portal vein using 20 mL ringer solution. The liver was placed on a silicon mat in the perfusion chamber exposing the hilum (Fig 1C). The portal vein, vena cava and hepatic artery (only in dNEVLP groups, Fig 1D) were connected to the perfusion circuit. The bile duct catheter tube was inserted into a pre-weighed collection tube to allow for free bile outflow. Cold ischemia time did not exceed 60 minutes. After connection to the perfusion circuit, the flow rates were slowly increased over a rewarming period of 15 minutes. T0 was set when full flows were reached. Subsequently the liver was perfused for six hours.

## Composition of the perfusate

Erythrocytes and plasma were separated by centrifugation at 4˚C and 3200 RPM for 15 minutes. The plasma phase was collected, and the buffy coat was withdrawn by suction. 10 mL of the erythrocyte concentrate were suspended in 35 mL of Dulbecco's Modified Eagle's Medium as used by Gassner et al. [23]. 5 mL of strain specific rat plasma were added generating a total perfusion volume of 50 mL with a calculated haematocrit of 20%. The perfusate was additionally supplemented with 1000 IE heparin and 12mM glycine.

## Measurement of biochemical markers and blood gas analysis

Perfusate samples were taken at the start, after 3 and 6 hours of perfusion and centrifuged at 3200 RPM and 4˚C for 10 minutes. The supernatant was collected for analysis. Alanine aminotransferase (ALT), aspartate aminotransferase (AST) and Urea were photometrically measured by *Labor Berlin–Charité Vivantes GmbH*. Additionally, blood gas analysis was performed on samples from the in- and outflowing perfusate (*ABL800 FLEX*, *Radiometer GmbH*, Berlin, Germany). Oxygen uptake was calculated according to *Tolboom et al.* [24]

## Tissue sampling and histological analysis

After perfusion livers were removed from the perfusion chamber and flushed with 20 mL of Ringer solution. At least four tissue samples were collected from each liver lobe and preserved in formaldehyde and at -80˚C for later histological analysis. Haematoxylin and eosin (H&E) staining (*AppliChem*, Darmstadt, Germany) was performed on 2 μm and 5 μm thick paraffin sections. A pathologist (R.A.) examined H&E-stained slices from a minimum of four different lobes from each perfused liver, blinded to the treatment groups. Levels of hepatocellular ballooning, loss of nucleus, and cellular fragmentation were assessed as markers for necrosis and sinusoidal dilatation was determined. Additionally, the hilar part of the extrahepatic bile duct, proximal to the catheter, was removed from the liver, frozen in liquid nitrogen and stored at -80˚C for later analysis. H&E staining was performed on 8 μm cryo sections. Terminal deoxynucleotidyl transferase dUTP nick end labeling (TUNEL) for special tissues (*F. Hoffmann-La Roche AG*, Basel, Switzerland) and 4′,6-diamidino-2-phenylindole (DAPI) staining (*Thermo Fisher Scientific Inc*., Waltham, Massachusetts, USA) was performed on 8 μm cryo sections according to the manufacturer's instructions. TUNEL-positive areas were measured using *NIH ImageJ* (Version: 2.0.0-rc-68/1.52f).

## Bile collection and analysis

The bile was continuously collected, weighed hourly, frozen in liquid nitrogen and then stored at -80˚ C for later analysis. In order to assess epithelial cell death in the extrahepatic bile duct, lactate dehydrogenase (LDH) and gamma glutamyltransferase (GGT) were determined in the collected bile. LDH was determined by photometric measurement (LDH activity assay kit, *Sigma-Aldrich Chemie GmbH* Munich, Germany) and GGT was determined by an ELISA (*Elabscience*, Houston, USA) according to the manufacturer's instructions.

## Statistical analysis

Data is presented as median in the text and as median and interquartile range in the tables. Categorical variables were analysed using the chi-squared test. After testing for normality using the Shapiro-Wilk test, group variables were analysed with two-way ANOVA or Kruskal-Wallis test and Bonferoni post-hoc test, accordingly. Data is presented as 95% confidence interval (CI). Overall, a p value < 0.05 was considered significant. Calculations were carried out using IBM SPSS Statistics Version 24.0 for macOS (*IBM Corp.*, Armonk, NY, USA). Graphs were generated using GraphPad Prism Version 8.11 (*GraphPad Software*, La Jolla, CA, USA).

# Results

## Stable perfusion conditions were achieved for all groups

Animal and liver weight did not differ significantly between all four groups (p = 0.47 and p = 0.34, respectively). Surgical procedures were performed in the same manner for all experiments apart from the arterial preparation in the dNEVLP groups. Cold ischemia time and macroscopic flush after explanation did not differ significantly. Initial pH was within physiological range and decreased to slightly acidic conditions in all experiments without significant differences ($pH_{T0}$ p = 0.22, $pH_{T3}$ p = 0.66, $pH_{T6}$ p = 0.22). Sodium, potassium and chloride stayed within physiological ranges in all four groups. Throughout perfusion, PV pressure did not significantly differ between all groups ($PVP_{T0}$ p = 0.26, $PVP_{T3}$ p = 0.63, $PVP_{T6}$ p = 0.87) and did not fall below or exceed the physiological range of 4–9 mmHg in all four groups throughout perfusion. Histopathology did not show relevant edema in any of the four groups after perfusion. Oxygen consumption remained high throughout perfusion in all groups without significant differences ($dNEVLP^{MP}$: $VO2_{T0}$ = 0.03 ml/min/g, $VO2_{T3}$ = 0.04 ml/min/g, $VO2_{T6}$ = 0.03 ml/min/g; $dNEVLP^{MH}$: $VO2_{T0}$ = 0.03 ml/min/g, $VO2_{T3}$ = 0.04 ml/min/g, $VO2_{T6}$ = 0.05 ml/min/g; $dNEVLP^{-M}$: $VO2_{T0}$ = 0.04 ml/min/g, $VO2_{T3}$ = 0.05 ml/min/g, $VO2_{T6}$ = 0.04 ml/min/g; sNEVLP: $VO2_{T0}$ = 0.02 ml/min/g, $VO2_{T3}$ = 0.04 ml/min/g, $VO2_{T6}$ = 0.03 ml/min/g).

## Administration of metamizole significantly decreased arterial pressure and increased bile production

Initial portovenous and arterial pressures did not significantly vary between all dNEVLP groups ($PVP_{T0}$ p = 0.09, $aP_{T0}$ p = 0.67, **Fig 2A and 2B**). In the $dNEVLP^{-M}$ group, the arterial pressure exceeded the physiological cutoff of 110 mmHg after three hours of perfusion leading to severely high pressures of up to 190 mmHg after six hours of perfusion. A significant decrease in arterial pressure was observed after both the hourly and the pressure dependent administration of metamizole in the metamizole treatment groups. This effectively prevented an increase of the arterial pressure above 110 mmHg ($aP_{T3}$ p = 0.007, $aP_{T4}$ p = 0.01, $aP_{T5}$ p = 0.02, $aP_{T6}$ p = 0.03, **Fig 2B**). Arterial pressures did not significantly differ between the two metamizole treatment groups.

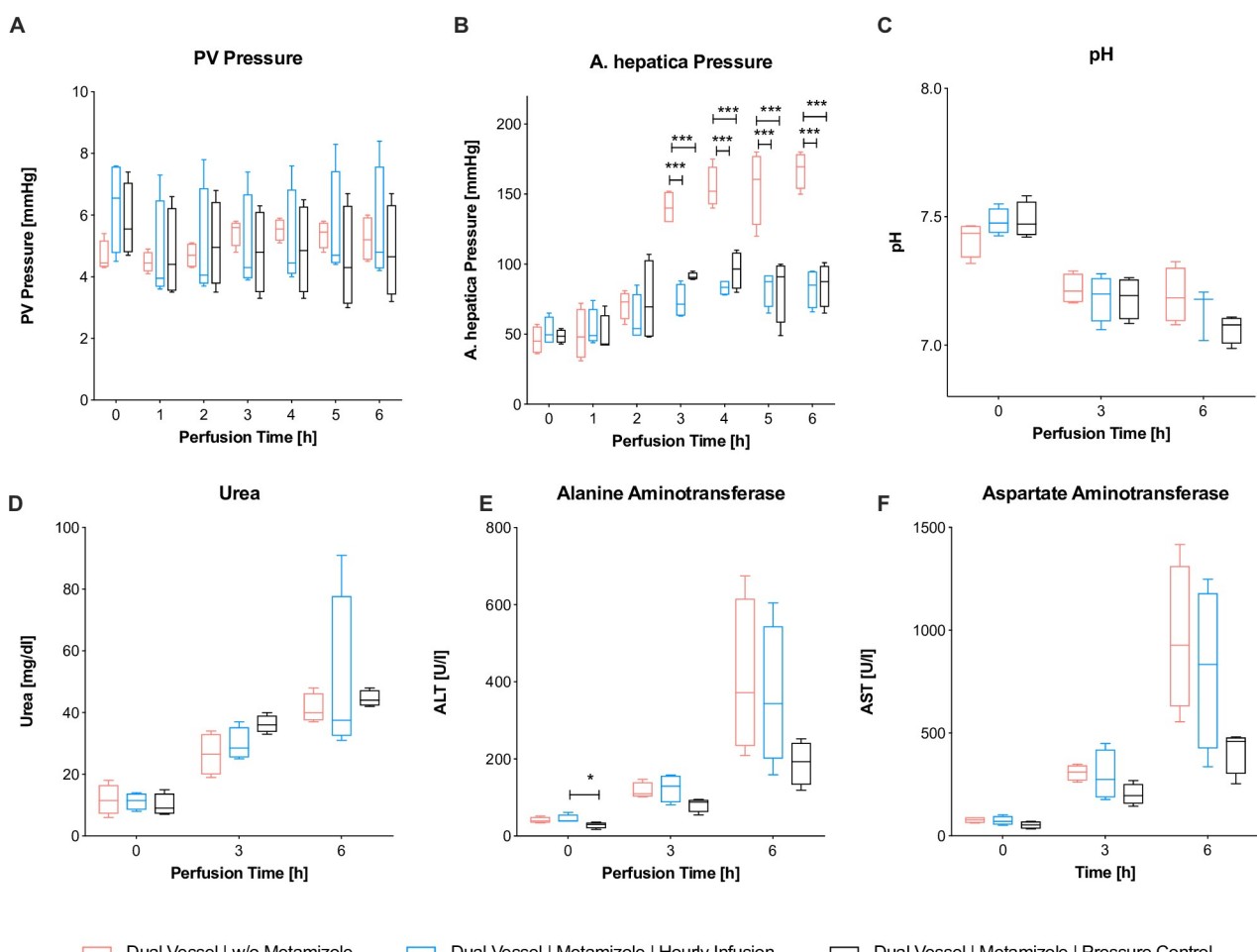

**Fig 2. Comparison of dNEVLP groups 1.** Comparison of the three dNEVLP groups: (A) portovenous pressure, (B) arterial pressure, (C) perfusate pH, (D) perfusate urea, (E) perfusate alanine aminotranferase, (F) perfusate aspartate aminotransferase. *** indicates p $\leq$ 0.001. Data shown as median and interquartile range.

Bile production significantly increased in the dNEVLP groups with metamizole application. In the second hour of perfusion bile production in the dNEVLP$^{MH}$ group was significantly higher than in the dNEVLP$^{MP}$ group, in which metamizole had not yet been administered, but not significantly higher than in the dNEVLP$^{-M}$ group (B$_{T2}$ p = 0.02, **Fig 3A**). In the fourth hour of perfusion, bile production in the dNEVLP$^{MH}$ group was significantly higher than in the dNEVLP$^{-M}$ group (B$_{T4}$ p = 0.04, **Fig 3A**). In the fifth and sixth hour of perfusion bile production in the dNEVLP$^{MP}$ group was significantly higher than in the dNEVLP$^{-M}$ group (B$_{T5}$ p = 0.04, B$_{T6}$ p = 0.03, **Fig 3A**).

## Administration of metamizole on demand (dNEVLP$^{MP}$) achieved lowest markers of liver and bile duct damage

Between all three dNEVLP groups perfusate urea levels and pH did not differ significantly throughout perfusion (U$_{T0}$ p = 0.79, U$_{T3}$ p = 0.1, U$_{T6}$ p = 0.32, pH$_{T0}$ p = 0.23, pH$_{T3}$ p = 0.78, pH$_{T6}$ p = 0.14, **Fig 2C and 2D**). In the dNEVLP$^{MH}$ and dNEVLP$^{MP}$ groups lactate levels after six hours of perfusion were lower than in the dNEVLP$^{-M}$ groups. However, differences did not reach statistical significance (L$_{T6}$ p = 0.14).

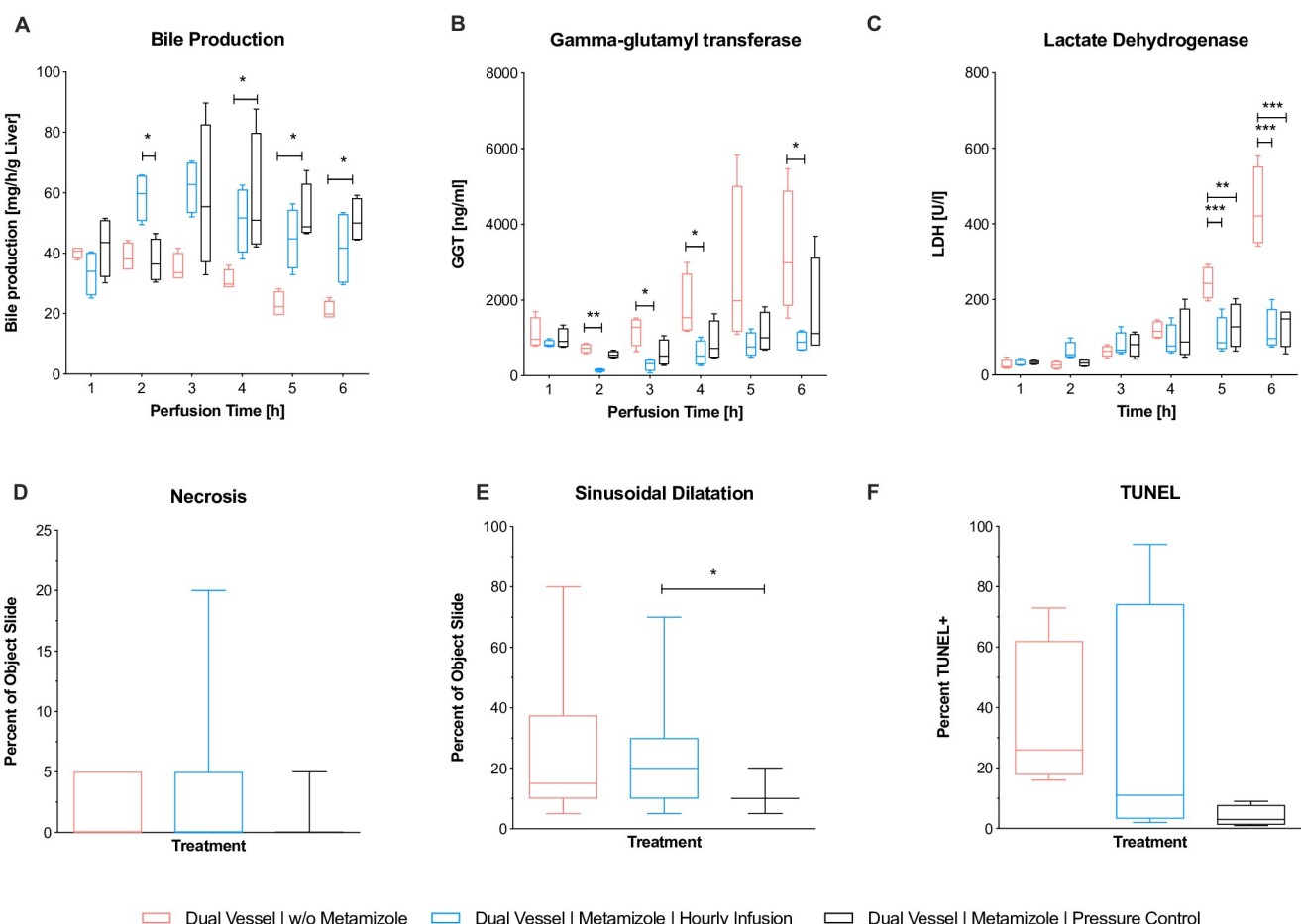

**Fig 3. Comparison of dNEVLP groups 2.** Comparison of the three dNEVLP groups: (A) amount of bile production, (B) bile gamma-glutamyl transferase, (C) lactate dehydrogenase within bile, (D) liver parenchyma necrosis, (E) liver parenchyma sinusoidal dilatation, (F) bile duct necrosis in TUNEL staining. * indicates p ≤ 0.05, ** p ≤ 0.01 and *** p = 0.001. Data shown as median and interquartile range.

Initial ALT levels were significantly lower in the dNEVLP$^{MP}$ group than in the dNEVLP$^{-M}$ group (ALT$_{T0}$ p = 0.048; **Fig 2E**). In the dNEVLP$^{MP}$ group, ALT and AST levels trended to be lower than in dNEVLP$^{MH}$ and dNEVLP$^{-M}$ groups throughout perfusion (**Fig 2E and 2F**)

HE staining showed considerably less sinusoidal dilatation in the dNEVLP$^{MP}$ group than in the dNEVLP$^{-M}$ group and significantly less sinusoidal dilatation than in the dNEVLP$^{MH}$ group after six hours of perfusion (p = 0.01, **Figs 3E and 4A–4C**). Tissue necrosis was also lowest in the dNEVLP$^{MP}$ group (**Figs 3D, 4A–4C**). However, no statistical significance could be shown (p = 0.9).

Measurement of bile LDH showed similar levels for all three dNEVLP groups during the first four hours of perfusion. After five and six hours of perfusion bile LDH levels were significantly lower in both metamizole treatment groups compared to the dNEVLP$^{-M}$ group (p = 0.02, **Fig 3C**).

Throughout perfusion bile GGT levels in the metamizole treatment groups were lower than in the non-treatment group. After two, three, four and six hours of perfusion, GGT levels in the dNEVLP$^{MH}$ group were significantly lower than in the dNEVLP$^{-M}$ group (**Fig 3B**). GGT levels did not differ significantly between the two metamizole treatment groups.

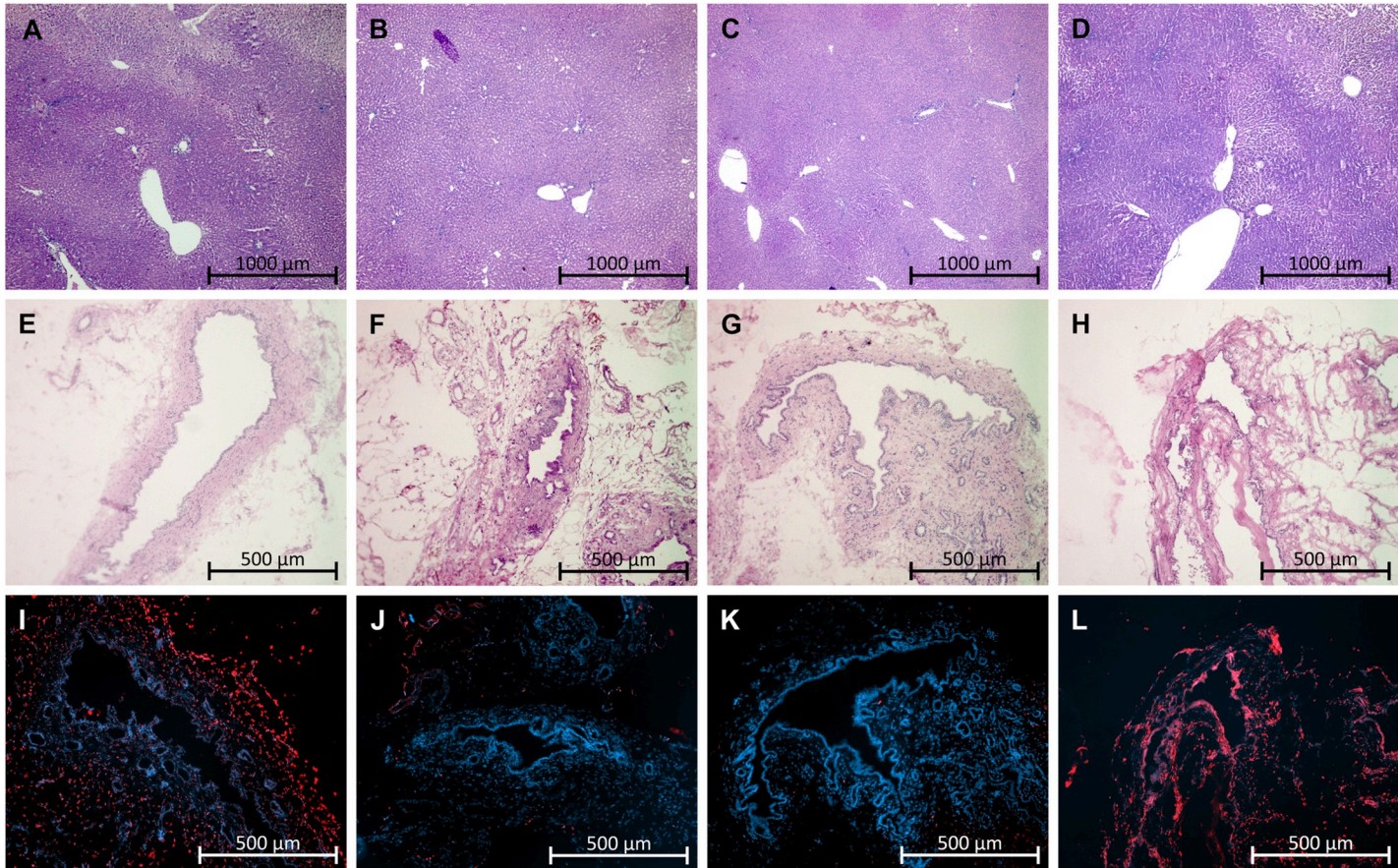

**Fig 4. Histopathology.** (A-D) HE staining of liver parenchyma, (E-H) HE staining of bile duct, (I-L) TUNEL & DAPI staining of bile duct, (A, E, I) dNEVLP$^{-P}$, (B, F, J) dNEVLP$^{MH}$, (C, G, K) dNEVLP$^{MP}$, (D, H, L) sNEVLP.

HE and TUNEL staining of the bile duct clearly showed lower necrosis in the dNEVLP$^{MP}$ group compared to the other two groups (**Figs 3F**, **4E–4G** and **4I–4K**). However, no statistical significance could be shown (p = 0.06).

## dNEVLP with administration of metamizole on demand (dNEVLP$^{MP}$) showed better liver function and lower markers of liver and bile duct damage, compared to sNEVLP

Between the two groups, perfusate pH and PV pressure did not show significant differences throughout perfusion (**Fig 5A and 5B**). Urea levels were higher in the dNEVLP$^{MP}$ group, but did not reach significance (**Fig 5C**). Lactate levels after six hours of perfusion were lower in the dNEVLP$^{MP}$ group than in the sNEVLP group even though differences did not reach statistical significance. Bile production was significantly higher in the dNEVLP$^{MP}$ group from three hours of perfusion on until the end (B$_{T4}$ p = 0.03, B$_{T5}$ p = 0.03, B$_{T6}$ p = 0.03, **Fig 6A**).

ALT levels were significantly lower in the dNEVLP$^{MP}$ group after three hours and still lower after six hours of perfusion (ALT$_{T3}$ p = 0.03; ALT$_{T6}$ p = 0.11; **Fig 5E**). AST levels in the dNEVLP$^{MP}$ group were lower after three and significantly lower after six hours of perfusion (AST$_{T3}$ p = 0.06, AST$_{T6}$ p = 0.03, **Fig 5D**).

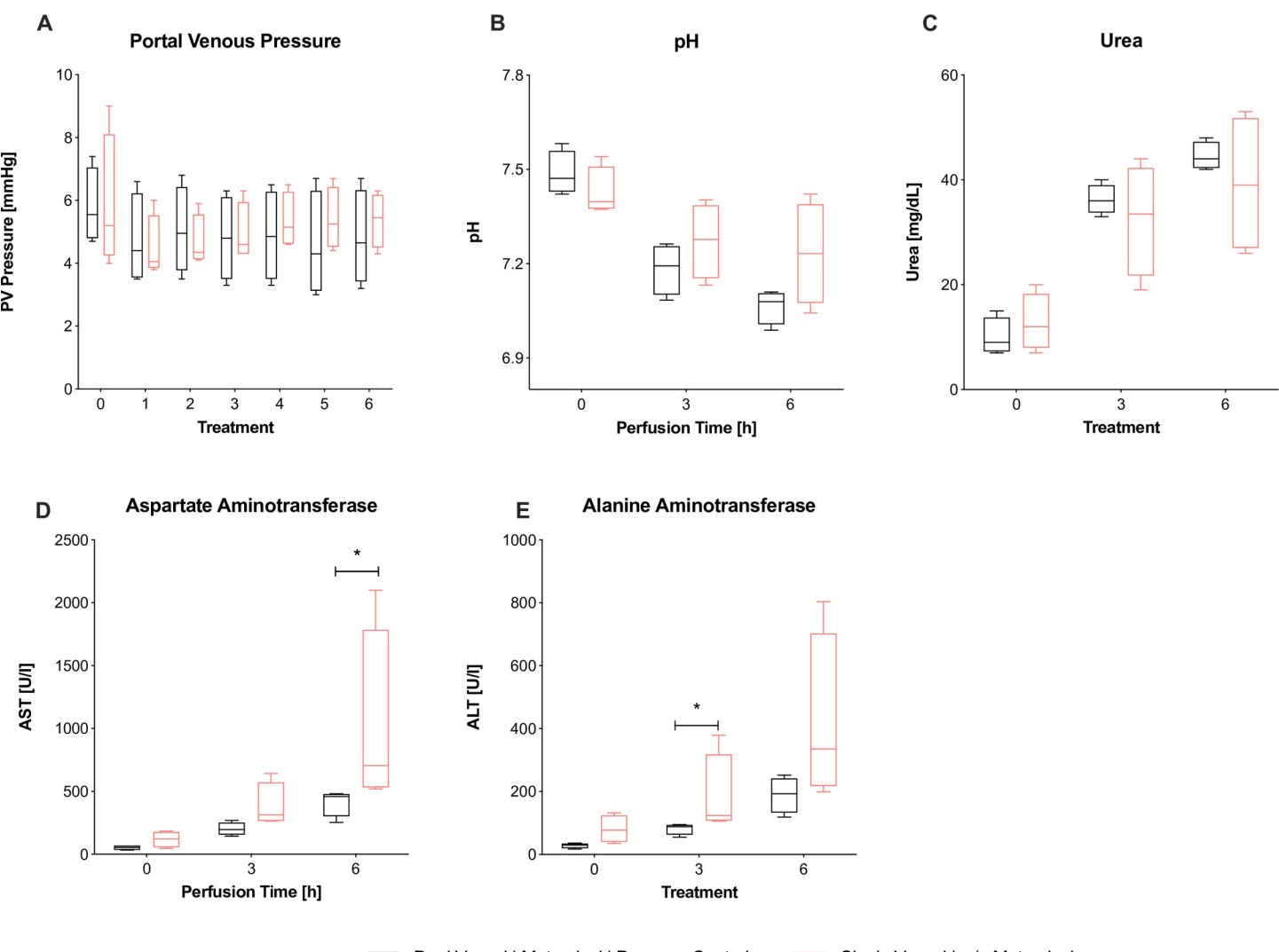

**Fig 5. Comparison of dNEVLP$^{MP}$ and sNEVLP groups 1.** Comparison of dNEVLP$^{MP}$ and sNEVLP: (A) portovenous pressure; (B) pH of perfusate, (C) urea within perfusate, (D) alanine aminotranferase within perfusate, (E) aspartate aminotransferase within perfusate. * indicates $p \leq 0.05$. Data shown as median and interquartile range.

H&E staining showed significantly lower sinusoidal dilatation and necrosis in the dNEVLP$^{MP}$ group after six hours of perfusion (necrosis $p = 0.02$, sinusoidal dilatation $p = <0.001$, **Figs 6D–6E, 4C and 4D**).

Bile LDH levels showed similar developments in both groups in the first four hours of perfusion. After five and six hours, LDH levels in the dNEVLP$^{MP}$ group were considerably lower than in the sNEVLP group. However, no statistical significance could be shown (**Fig 6C**). As well, GGT levels trended to be lower in the dNEVLP$^{MP}$ group from three hours of perfusion on (**Fig 6B**).

H&E and TUNEL staining of the bile duct showed significantly lower necrosis in the dNEVLP$^{MP}$ group ($p = 0.03$, **Figs 6F, 4G–4H and 4K–4L**).

## Discussion

Normothermic ex vivo liver perfusion (NEVLP) is regarded as a beneficial alternative to static cold storage in liver transplantation, especially when using marginal liver grafts. Small animal

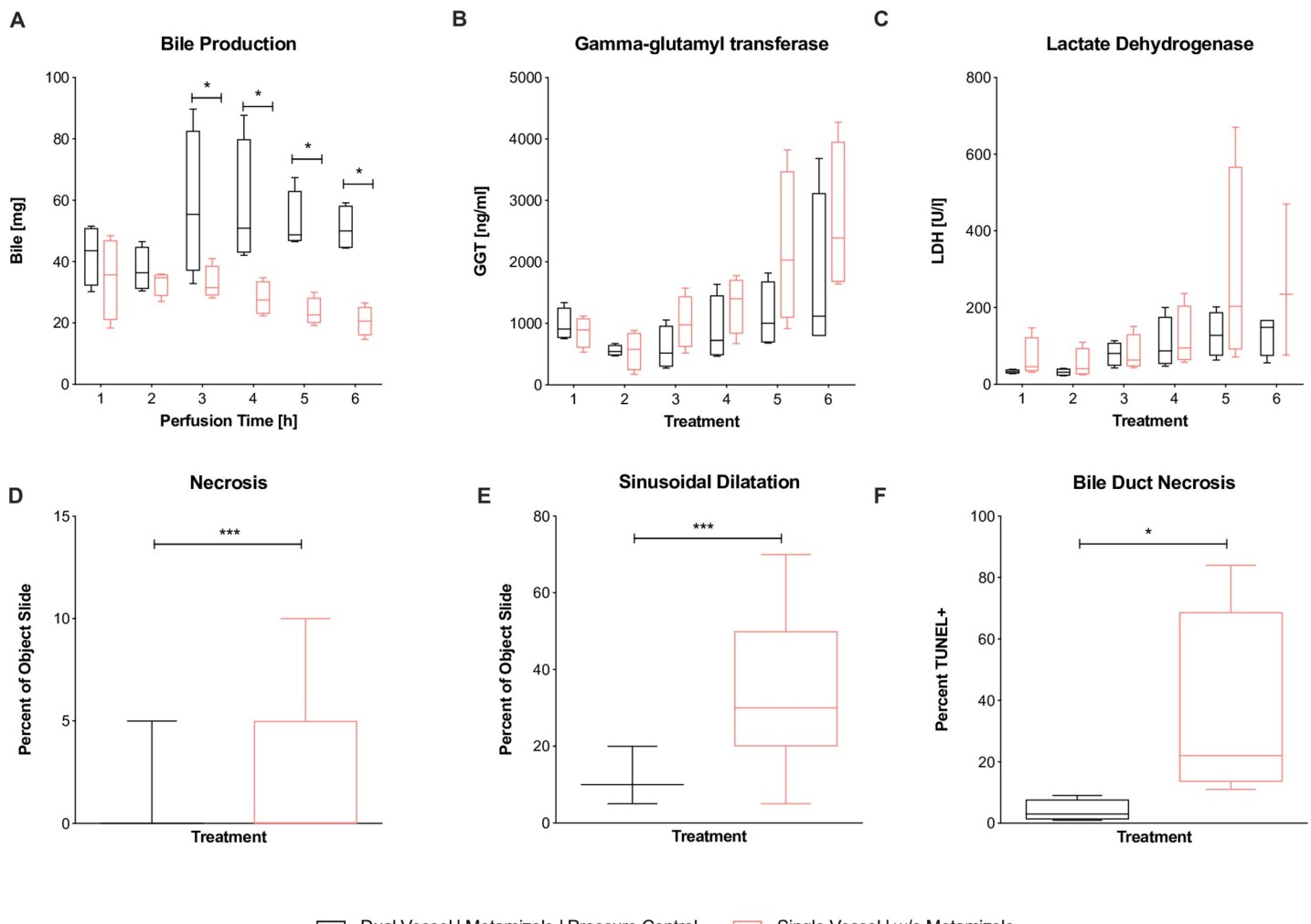

**Fig 6. Comparison of dNEVLP^MP and sNEVLP groups 2.** Comparison of dNEVLP^MP and sNEVLP: (A) amount of bile production, (B) gamma-glutamyl transferase within bile, (C) lactate dehydrogenase within bile, (D) liver parenchyma necrosis, (E) liver parenchyma sinusoidal dilatation, (F) bile duct necrosis in TUNEL staining. * indicates p ≤ 0.05 and *** p = 0.001. Data shown as median and interquartile range.

NEVLP models are needed to foster the development of strategies for organ preservation and reconditioning. In current literature, laboratory-scaled NEVLP models usually utilize single vessel perfusion, which means the organ is only perfused through the PV. *Tolboom et al.* could show that single vessel normothermic liver perfusion is a feasible strategy for organ preservation of rat livers with high survival rates after transplantation [24]. However, when it comes to metabolic reconditioning, there are many arguments for a dual vessel approach: 9–12% of rat liver parenchyma is supplied by the arterial blood flow only [25]. The extrahepatic bile duct obtains at least 49% of its blood supply exclusively from the HA and its epithelial cells are known to be especially vulnerable to ischemia [26, 27]. *Mora et al.* showed, that in porcine extracorporeal liver perfusion as a method to provide temporary liver support for patients with severe liver failure, dually perfused livers performed better than such solely perfused trough the PV [28]. Generally, a dual vessel perfusion model corresponds to the physiological situation better than a single vessel perfusion model. Most importantly, it more directly reflects the clinical situation, since NEVLP devices for human liver grafts also employ dual vessel perfusion [18–20].

Despite these considerations, there are only a few studies reporting on dual vessel small animal NEVLP. *Op den Dries et al.* and *Schlegel et al.* presented dual vessel rat NEVLP models in their recent works, realizing perfusion periods of three and four ours, respectively [15, 29]. However, for ex vivo organ reconditioning, perfusion periods of more than four hours may be necessary to reflect the clinical situation, where perfusion time frequently exceeds such durations. Moreover, in our opinion, longer perfusion periods could be necessary to effectively conduct pharmacological organ reconditioning.

Although there are several arguments for dual vessel NEVLP, single vessel small animal NEVLP models are still commonly used in basic research. However, there is no evidence in literature for the equivalence of the single vessel and the dual vessel approach or even the superiority of one over the other. *t´Hart et al.* compared single and dual vessel NEVLP but were unable to accomplish perfusion periods of longer than 90 minutes without witnessing a severe escalation of arterial perfusion pressure and transaminase levels [30]. Reasons for that could include the lack of a dialysis circuit and the utilization of a cell free perfusate, as previous findings of our work group show [31, 23]. *Brüggenwirth et al.* recently published a comparative study on dual and single vessel end-ischemic normothermic reperfusion of the rat liver following six hours of cold storage and one hour of subnormoythermic reperfusion [32]. They showed similar outcomes for dual and single vessel perfused livers. In our opinion, short end-ischemic reperfusion after prolonged cold storage might be a feasible strategy for organ preservation but not the right option for metabolic reconditioning of rat liver grafts.

The first aim of this work was to develop a dual vessel NEVLP rat model that would maintain near-to-physiological conditions for a perfusion period of six hours. The development of such a model comes with many difficulties. The most serious problem we observed was a severe increase of vascular resistance in the arterial flow area. We attributed this phenomenon to progressive vasospasm in the small arterial vessels of the rat liver and assumed that sufficient vasodilatation would be necessary for successful dual vessel NEVLP. We identified metamizole as a possible agent to accomplish that. *Kaya et al.* have shown that metamizole is able to effectively ameliorate arterial vasospasm of the hepatic artery, when applied topical [22]. We investigated, whether the direct administration of metamizole into the HA could effectively decrease arterial vascular resistance in the liver and ensure sufficient vasodilatation. In many countries, including Germany, metamizole is inexpensive, widely available and therefore often used in animal research and veterinary medicine. It has a low risk of causing acute liver failure and has been shown to have no toxic effects on hepatocytes [33, 34]. Agranulocytosis, which is a known severe adverse effect of metamizole treatment, has led to its ban in several countries. Although we do not propose using metamizole for clinical NEVLP, the risk of myelotoxic effects during leukocyte-free NEVLP should be low. Since metamizole has not been shown to accumulate in the liver and the liver would be flushed before transplantation, the risk of adverse effects after transplantation should also be low.

Our results confirm our previous observation of increasing vascular resistance occurring from three to four hours of perfusion onward. As presumed, the subsequent unphysiologically high arterial pressures resulted in poor perfusion outcome, as parameters of liver and bile duct damage were considerably elevated. Interestingly, the high arterial pressures did not lead to relevant development of edema. We could show, that the application of metamizole into the HA ensured sufficient vasodilatation and kept arterial pressures in the physiological range. This led to better organ preservation as both metamizole groups showed lower markers of liver damage as well as better results in the histopathology of both liver and bile duct tissue. Notably, the application metamizole significantly increased bile production and lowered levels of bile duct damage markers.

The comparison between the two metamizole groups indicated, that the administration of metamizole on demand–as opposed to a static hourly administration–was altogether more beneficial for the organs. Interestingly, the hourly administration of metamizole even led to significantly more sinusoidal dilatation than the pressure dependent administration on demand. This suggests, that metamizole itself does not have an intrinsic positive effect on the liver but develops its beneficial effect through vasodilatation, when vasodilatation is needed.

Furthermore, the on-demand-protocol accomplished more stable perfusion conditions, as results in this group were the most consistent ones between the three dNEVLP groups.

Our results show that dual vessel NEVLP is only beneficial if sufficient vasodilatation is performed during perfusion. We propose Metamizole as one possible agent to accomplish that. However, other vasodilators (e.g. Epoprostenol or Verapamil) might show similar results [35].

The second aim of this work was the evaluation of our newly developed dNEVLP system by comparing it to our well-established sNEVLP system. Our results indicate that dNEVLP leads to lower liver damage and better liver function. Although oxygen uptake did not show a significant difference, histological architecture was better preserved, as necrosis and sinusoidal dilatation in the liver parenchyma were significantly lower. We attribute this not only to the improved perfusion of areas that are supplied only by the HA, but also to a general improvement of the microcirculation of the liver. Histopathology also showed significantly better preservation of the extra hepatic bile duct by dNEVLP compared to sNEVLP. Since the extra hepatic bile duct obtains a great part of its blood supply only from the hepatic artery, these results seem coherent. Chemical markers of bile duct epithelial damage show the same trend, but results did not reach statistical significance. One reason for this is the high standard deviation of the results in the sNEVLP group. Another reason might be, that bile duct damage in the sNEVLP group was more frequently located in the deeper layers of the bile duct than in the epithelium, as histopathology showed. Again, it is important to mention that dNEVLP achieved more stable perfusion conditions, as results were more consistent than in the sNEVLP group.

Our results suggest, that dNEVLP with on demand application of metamizole for vasodilatation leads to superior organ preservation after six hours of perfusion compared to sNEVLP in our miniaturized rat liver perfusion system. Judging from our oxygen consumption analysis, liver function tests, and histological analyses, the organs were well perfused and viable throughout the entire perfusion period. Moreover, all livers in our experiments met the clinical criteria for viability and transplantability as established by *Mergental et al.* (bile production, stable flow rates and homogenous perfusion) [36]. However, these criteria were established for human livers and thus have limited validity in the assessment of rat livers. It remains to be seen, if dually perfused rat livers also perform better after transplantation. This is currently being investigated in our work group.

In conclusion, we here present a dual vessel small animal NEVLP model. We introduce metamizole as a potent agent to mitigate arterial vasospasm, thus allowing for perfusion periods of six hours and more. Furthermore, we present a comparison between single and dual vessel NEVLP in a small animal model. We show, that dNEVLP with sufficient vasodilatation by metamizole application on demand seems to be superior to sNEVLP in terms of organ preservation.

## Supporting information

**S1 File.**
(DOCX)

## Acknowledgments

The Authors would like to thank Steffen Lippert, Kirsten Führer and Dietrich Polenz for their instruction and support in the methods.

## Author Contributions

**Conceptualization:** Felix Claussen, Joseph M. G. V. Gassner, Simon Moosburner, Maximilian Nösser, Julian Pohl, Anja Reutzel-Selke, Igor M. Sauer, Nathanael Raschzok.

**Data curation:** Felix Claussen, Simon Moosburner, David Wyrwal, Maximilian Nösser, Lara Wegener, Julian Pohl, Ruza Arsenic.

**Formal analysis:** Felix Claussen, Simon Moosburner, David Wyrwal, Anja Reutzel-Selke.

**Funding acquisition:** Anja Reutzel-Selke, Nathanael Raschzok.

**Methodology:** Peter Tang, Julian Pohl, Ruza Arsenic.

**Project administration:** Johann Pratschke, Igor M. Sauer.

**Resources:** David Wyrwal, Peter Tang, Lara Wegener.

**Supervision:** Joseph M. G. V. Gassner, Peter Tang, Anja Reutzel-Selke, Johann Pratschke, Igor M. Sauer, Nathanael Raschzok.

**Validation:** Simon Moosburner, Peter Tang, Anja Reutzel-Selke, Ruza Arsenic, Johann Pratschke.

**Visualization:** Felix Claussen, Simon Moosburner.

**Writing – original draft:** Felix Claussen.

**Writing – review & editing:** Felix Claussen, Joseph M. G. V. Gassner, Johann Pratschke, Igor M. Sauer, Nathanael Raschzok.

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
