## [Decision Letter · Decision Letter 0]

30 Mar 2020

PONE-D-20-06139

Dual versus single vessel normorthermic ex vivo perfusion of rat liver grafts using metamizole for vasodilatation.

PLOS ONE

Dear Prof. Dr. Sauer,

Thank you for submitting your manuscript to PLOS ONE. After careful consideration, we feel that it has merit but does not fully meet PLOS ONE’s publication criteria as it currently stands. Therefore, we invite you to submit a revised version of the manuscript that addresses the points raised during the review process.

We would appreciate receiving your revised manuscript by May 14 2020 11:59PM. To enhance the reproducibility of your results, we recommend that if applicable you deposit your laboratory protocols in protocols.io, where a protocol can be assigned its own identifier (DOI) such that it can be cited independently in the future. For instructions see: http://journals.plos.org/plosone/s/submission-guidelines#loc-laboratory-protocols

We look forward to receiving your revised manuscript.

Kind regards,

Michael Bader

Academic Editor

PLOS ONE

Journal Requirements:

2. Thank you for stating the following in your manuscript:

"This work was funded by institutional financial support of the Charité – Universitätsmedizin Berlin, the German Research Foundation and by a Kickbox Seed Grant of the Einstein Center for Regenerative Therapies. Nathanael Raschzok is fellow of the BIH Charité Clinician Scientist Program funded by the Charité – Universitätsmedizin Berlin and the Berlin Institute of Health."

Reviewers' comments:

Reviewer's Responses to Questions

**Comments to the Author**

1. Is the manuscript technically sound, and do the data support the conclusions?

Reviewer #1: Yes

Reviewer #2: Yes

Reviewer #3: Partly

2. Has the statistical analysis been performed appropriately and rigorously? 

Reviewer #1: Yes

Reviewer #2: Yes

Reviewer #3: Yes

3. Have the authors made all data underlying the findings in their manuscript fully available?

Reviewer #1: Yes

Reviewer #2: Yes

Reviewer #3: Yes

4. Is the manuscript presented in an intelligible fashion and written in standard English?

Reviewer #1: Yes

Reviewer #2: Yes

Reviewer #3: Yes

5. Review Comments to the Author

Reviewer #1: This study was performed in a rat model of ex vivo liver perfusion using dual versus single vessel perfusion. The authors evaluate these effects after the use of metamizole.

Although the authors showed a relationship between the use of metamizole and the protection against the arterial vasospasm with is important because one of the main complications associated with liver transplantation is related to bile duct ischemia.

Despite these, I sent my comments:

The heterogeneous etiologies for liver transplantation and methodology were problematic. In this study, the time of cold ischemia and warm ischemia is very short. One of the main problems of this work is that it is barely reproducible since cannulating a rat's hepatic artery is technically demanding. I suggest placing an image in the article.

During this study, the liver was flushed via the aorta and the portal vein with HTK solution supplemented with glycine. Given the variability of liver preservation solutions, will this finding be transversal to other preservation solutions? UW or Celsior?

The underlying molecular mechanism was not well suggested in this study. What is the effect in the mitochondrial function, the principal cellular source of energy? For example, the content of ATP?

How do you think it could be used in liver transplantation?

Reviewer #2: This paper provides a novel technique at delivering metamizole through the hepatic artery during single or dual vessel ex vivo perfusion of the rat liver to improve perfusion outcomes. Those livers perfused with metamizole, whether through pressure control or hourly administration, appeared to have less ischemic injury as evidenced by greater urea production and lower AST and ALT levels as well as more physiologic pH values. My critique of this paper is that this medication appears to be banned in the United States for human use by the FDA. Therefore, I recommend a major revision of your conclusion portion of the manuscript. Namely, I caution you in your statement that "metamizole is a widely used drug in human medicine" as well as your notations qualifing its limited risks of agranulocytosis. The main limitation of this paper, while well designed, is that the drug used has limited clinical utility.

Reviewer #3: In this study, the authors present a comparison between dual vessel vs. single vessel normothermic liver perfusion in a small animal model. The authors also focused on a known perfusion pressure issue associated with hepatic artery perfusion by introducing metamizole as an agent to mitigate arterial vasospasm. They hypothesized based on previous research, that since metamizole has been shown to decrease vasospasm in the rat femoral artery, that these findings may also decrease pressure related issues during hepatic artery perfusion. As a result, the authors conclude dual vessel liver perfusion to be superior to single vessel perfusion in the presence metamizole for use in arterial pressure control for perfusions that last 6 or more hours.

This is a well-designed and rigorous study with one main exception: The work is persuasive for the use of metamizole improving arterial pressures during dual vessel perfusion in small animal models; however there appears need of significant data and likely another experimental group to back up the claims that dual vessel perfusion is superior to single vessel perfusion in this small animal model. There are also some questions regarding the methods employed in liver recovery and perfusion which may need clarifications or corrections.

It is unclear as to how a definitive conclusion can be reached that dual vessel perfused livers perform better than single vessel perfusion. First, unless I am missing it there was no single vessel perfusion with metamizole group, which makes comparisons difficult. It would have been a better comparison if single vs dual perfusion groups compared without metamizole, as it would be an apples to apples comparison. Still, a question that rises is if metamizole would have had similar beneficial effects in single vessel perfusion, and if the results would then be comparable to dual+ metamizole group. In addition, oxygen consumption and oxygen saturation levels at inlet and effluent would be important to evaluate, since providing oxygen via both vessels is a key benefit of dual perfusion based on human liver perfusion literature. Supplemental data shows some lactate clearance, but it is unclear if the dialysis cartridge used removes lactate from the perfusate. Other indicators that come to mind are edema and ATP levels; another suggestion is to use some of the clinical perfusion viability criteria already used, such as the one published by Mergental et al to compare the viability of the perfused grafts.

A more technical concern that could potentially influence the perfusion metrics and overall outcome was ligating the SHVC and cannulating the IVC. This can restrict the outflow of the perfusate and can increase the overall pressure within the liver. Research has shown that even partial occlusion of veins in the liver can lead to diffused hepatic congestion and enlargement (ie. sinusoidal thrombosis). This effect may have several different downstream consequences, all of which could alter the evaluation of the grafts. The results of the graft assessment given (pressure (both arterial and portal), histological analysis, biochemical markers ALT/AST), could be influenced by complications arising from the methodology of the experimental design itself. Have the authors considered such issues, and if so how did they avoid it affecting their results?

The final technical concern is the flow rates that were used in this study. For dual vessel perfusion the flow was set to 1.1mL/min/g (paragraph 201-202) generating a flow of 1ml/min/g liver through the portal and 0.1 ml/min/g liver through the hepatic artery. These flow rates are very low, compared with other papers in literature which listed the arterial flow from 0.21 +/- 0.02 to 3.5 +/- 0.2 mL/min/g liver, and portal flow 1.53 +/- 0.19 to 32.1 +/- 1.6 mL/min/g liver. The flow rates stated in the manuscript are consistent with mouse liver perfusions. With such low flow rates, the concern is about the availability of oxygenated perfusate to the liver and if the flow rates were high enough to fully oxygenate the organ. Without data on the oxygen consumption and saturation values, it is not possible to ascertain if the organ was oxygenated sufficiently.

Minor issues in the manuscript:

1. 1 mL Lactated Ringers that is supplemented with 500 IE Heparin is injected into the IVC. This concentration of Heparin in rats weighing between 280-350g will subsequently euthanize the rat exposing the liver to warm ischemic time (WIT). The portal and aorta are then cannulated and flushed after with 20 mL of 4C HTK, however the liver is still within the body cavity, and in the subjected to room temperature while the hepatic artery SHVC, right suprarenal vein, esophageal veins are ligated and the IVC is cannulated. Even though the WIT time is likely minimal, it should be clarified in the manuscript.

2. Average initial pressure was 5.65 mmHg. This pressure seems awfully high given that the temperature of the liver is close to 4C at the time of connection? This could cause damage to the endothelial layer within the vasculature of the liver and subsequently cause further complications during the length of perfusion.

3. There are various spelling and grammatical errors throughout the manuscript, which could use a thorough review by the authors.

6. PLOS authors have the option to publish the peer review history of their article (what does this mean?). If published, this will include your full peer review and any attached files.

Reviewer #1: No

Reviewer #2: Yes: Corey Eymard

Reviewer #3: No

---

## [Author Response · Author response to Decision Letter 0]

14 May 2020

Dear Dr. Heber,

Thank you very much for giving us the opportunity to submit a newly revised version of our manuscript. We hereby submit our revised manuscript as clean copy (Claussen_Manuscript_PlosOne_R1.docx) and additional copy with the changes being done highlighted (Claussen_Manuscript_PlosOne_R1_marked.docx). 

Please find below our specific answers to the questions and suggestions of the reviewers as a point-to-point response. We hope that our manuscript is now appropriate for publication in PLOS ONE.

Yours sincerely,

Reviewer #1

“The heterogeneous etiologies for liver transplantation and methodology were problematic. In this study, the time of cold ischemia and warm ischemia is very short. One of the main problems of this work is that it is barely reproducible since cannulating a rat's hepatic artery is technically demanding. I suggest placing an image in the article.”

Thank you for your suggestions. It is true that the cannulation of a rat’s hepatic artery is technically very demanding. We therefore developed a technique that makes this task considerably easier for the veterinary surgeon. Instead of directly cannulating the hepatic artery, the surgeon will cannulate the celiac trunk via a patch that is cut out of the aorta and then push the cannula forward into the hepatic artery. This ensures a much better and more secure grip on the vessel and makes the whole procedure much easier and quicker. Employing this technique, it does not take the veterinary surgeon more than 5-10 minutes to cannulate the hepatic artery, which considerably shortens warm ischemia time in our experiments (max. 15 minutes). Speaking from our experience, it will take an inexperienced trainee between 10 and 15 surgeries to securely learn this technique. 

To keep the cold ischemia time (i.e. time between cold flushing of the liver and connection to the perfusion circuit) short, it is important to have the perfusion circuit primed as soon as the liver is procured. If everything has been prepared thoroughly, a CIT of 60 minutes is practicable.

The cannulation of the hepatic artery is definitely one of the more challenging steps of our perfusion experiments. However, with the right technique, a decent amount of practice and thorough preparation, a WIT of 15 minutes and a CIT of 60 minutes will be reproducible.

We have adjusted the “Surgical procedures” part of our manuscript to make this more apparent and thank you for your question to clarify this.

“Methods” part, p. 10: “In the dNEVLP groups the hepatic artery was cannulated through an aortic patch (Fig. 1C/D).”

We have also included two pictures of the liver lying in the perfusion chamber, in which the cannulated artery (yellow cannula) and the aortal patch can be seen (Fig1 C, D).

“During this study, the liver was flushed via the aorta and the portal vein with HTK solution supplemented with glycine. Given the variability of liver preservation solutions, will this finding be transversal to other preservation solutions? UW or Celsior? The underlying molecular mechanism was not well suggested in this study. What is the effect in the mitochondrial function, the principal cellular source of energy? For example, the content of ATP? How do you think it could be used in liver transplantation?”

Glycine has been shown to have cytoprotective effects on hepatocytes as well as to ameliorate Kupffer cell activation in the liver (1, 2). These effects are mainly achieved by temporary hyperpolarization through glycine-gated chloride channels (3, 4). We assume that this mechanism and therefore the beneficial effects of glycine are independent of the preservation solution and would work in a similar way in UW or Celsior solution. We decided to use HTK instead of the other available preservation solutions because it is the gold standard for liver and kidney static cold storage in Germany.

The metabolic mechanisms of glycine treatment have extensively been investigated elsewhere (3, 4). Furthermore, the underlying mechanisms and beneficial effects of glycine treatment on ex vivo perfused livers have been investigated and discussed by our work group in previous publications (5). Analyzing the molecular mechanisms of glycin as part of our perfusion protocol was not an aim of this work and we relied on these previous findings when we chose to use glycine as a supplement to our preservation solution and perfusion medium.

HTK-N, a modified HTK solution including glycine, is already in clinical use and has been shown to have beneficial effects on organ preservation (6). 

Given the fact, that the aim of our work was to develop a small animal model for dual-vessel normothermic liver machine perfusion, which should be used for preclinical studies in a field with already well-established and well-working clinical systems, we did not aim to develop a solution that is immediately transferrable to clinical practice. We added this to the Discussion section. This is stated several times in the Introduction and Discussion section, e.g. page 7 (“In order to develop and investigate organ recovery strategies based on NEVLP, animal models are needed. … We developed a dual vessel normothermic ex vivo liver perfusion (dNEVLP) model for rat livers …”) or page 19 (“The first aim of this work was to develop a dual vessel NEVLP rat model…”). We also added this specific point to the Conclusion of the abstract (“Our miniaturized dNEVLP system enables normothermic dual vessel rat liver perfusion.”).

Reviewer #2

“My critique of this paper is that this medication appears to be banned in the United States for human use by the FDA. Therefore, I recommend a major revision of your conclusion portion of the manuscript. Namely, I caution you in your statement that "metamizole is a widely used drug in human medicine" as well as your notations qualifing its limited risks of agranulocytosis. The main limitation of this paper, while well designed, is that the drug used has limited clinical utility.”

Thank you very much for your remarks on our manuscript and for your appreciation of our study design. As requested, we removed the statement that "metamizole is a widely used drug in human medicine" and our statement regarding the limited risks of agranulotcytosis. Instead, we emphasize in the revised version of our Conclusion section that the aim of our work was to develop a dual-vessel model for normothermic rat liver perfusion for basic, preclinical research. 

The Discussion section, p. 20 was modified from “However, agranulocytosis has only been observed very rarely and metamizole is still regarded as a safe drug in literature and for example in regular use in Germany. Also, when using Metamizole in our NEVLP model, the risk of agranulocytosis should be very low because the perfusion circuit is leukocyte-free.” to “Although we do not propose using metamizole for clinical NEVLP, the risk of myelotoxic effects during leukocyte-free NEVLP should be low. Since metamizole has not been shown to accumulate in the liver and the liver would be flushed before transplantation, the risk of adverse effects after transplantation should also be low.”

We showed that sufficient vasodilatation is key to successful dual-vessel rat liver perfusion. We propose metamizole as a feasible agent to accomplish this. If there are specific concerns regarding the use of metamizole due to availability or other reasons, other vasodilators may be used. We altered our manuscript to clarify this. (“Discussion” part, p. 21: “Our results show that dual vessel NEVLP is only beneficial if sufficient vasodilatation is performed during perfusion. We propose Metamizole as one possible agent to accomplish that. However, other vasodilators (e.g. Epoprostenol or Verapamil) might show similar results.”)

Reviewer #3

“It is unclear as to how a definitive conclusion can be reached that dual vessel perfused livers perform better than single vessel perfusion. First, unless I am missing it there was no single vessel perfusion with metamizole group, which makes comparisons difficult. It would have been a better comparison if single vs dual perfusion groups compared without metamizole, as it would be an apples to apples comparison. Still, a question that rises is if metamizole would have had similar beneficial effects in single vessel perfusion, and if the results would then be comparable to dual+ metamizole group. “

The aim of this work was to develop a dual vessel perfusion model for rat liver grafts that reflects the in vivo situation of portal and arterial blood supply to the liver, and that would work as well as or even better than the commonly used single vessel model. We started of by performing dual vessel perfusion in analogy to our single vessel perfusion (Dual vessel w/o metamizole). We observed potential beneficial effects of the dual vessel approach e.g. lower levels of transaminases up to four hours of perfusion and higher bile production throughout perfusion compared to our single vessel perfusion experiments. However, these potential beneficial effects were outweighed by negative results, e.g. higher transaminase levels after six hours of perfusion, higher or similarly high levels of bile duct damage especially from four hours of perfusion onward and no improvement in tissue preservation/histopathology. We attributed these negative results to damages done by unphysiologically high arterial pressures that occurred from after three to four hours of perfusion.

From our point of view it was crucial to ensure near to physiological conditions including physiological arterial pressures before drawing a comparison between the two perfusion models. A comparison between dual vessel NEVLP w/o vasodilation and single vessel NEVLP would have been a comparison between a well-established and well working perfusion model (single vessel NEVLP) and an unready new perfusion model that was unable to sustain near-to-physiological conditions. In our opinion such a comparison would be of no consequence. 

The question, whether a single vessel NEVLP + metamizole group would be necessary to draw a more direct comparison and investigate similar beneficial metamizole effects on single vessel perfusion is understandable and we ourselves have considered including such a group in this work. However, in our opinion, there are several reasons that make a single vessel NEVLP + metamizole group unnecessary:

Regarding the potential positive effects of metamizole, our results show that the use of metamizole for vasodilatation is beneficial in dual vessel perfusion. However, the application mode plays an important role. Overall, the on-demand protocol showed better and more stable results than the static hourly protocol. Static hourly metamizole application led to higher levels of transaminases and worse histopathology. This indicates that metamizole itself does not have an absolute positive effect on the perfused liver, but should only be used, when vasodilatation is required. A single vessel perfusion model does not require vasodilation. In our experiments metamizole application did not have an effect on vascular resistance and pressure in the portovenous flow area. The use of a vasodilator like metamizole does not seem logical in such a model and we do not think that the addition of metamizole to our single vessel NEVLP model would have a beneficial effect. We altered the manuscript accordingly (“Discussion” part, p. 21: “This suggests, that metamizole itself does not have an absolute positive affects on the liver but develops its beneficial effect through vasodilatation, when vasodilatation is needed.”)

Regarding the comparability of a sNEVLP + metamizole group and the dNEVLP + metamizole on demand group, we have two concerns. The first one is the application route. Establishing our dual vessel perfusion model we found that vasodilatation was most sufficient, when the agent was administered directly into the hepatic artery. The application into both vessels usually led to insufficient vasodilatation and poorer perfusion outcome. We therefore decided to administer metamizole directly into the hepatic artery in all experiments of the two metamizole groups. Naturally, this would not be possible in our single vessel NEVLP model, since there is no artery. In a sNEVLP + metamizole group metamizole would thus have to be administered into the portal vein. This would lead to considerably higher concentrations of metamizole in the portovenous flow area than after injection into the heaptic artery, where the metamizole bolus is diluted by the perfusate before reaching the portal vein. An alternative approach would be to administer the bolus into the effluent of the liver, thus mimicking the metamizole outflow after arterial injection. However, the bolus would be diluted instantly and metamizole concentrations in the portal vein would likely be too low to have any effect. In our opinion, neither of the two strategies would improve the quality of the comparison between the single vessel NEVLP model and the dual vessel NEVLP + metamizole model. 

The second concern regards the scheme of application. In our optimized dual vessel NEVLP model we perform vasodilatation by on-demand metamizole application at a certain arterial pressure cut-off at different points in time. As mentioned above, in a single vessel NEVLP model there is no demand for vasodilatation and therefor no such cut-off. In a single vessel NEVLP + metamizole group metamizole would thus have to be administered at completely random points in time without any underlying principle. This does not seem reasonable and would, again, not improve the quality of the comparison.

We agree that we cannot completely rule out any effects metamizole might have besides its properties as a vasodilator. However, if the vasodilator used does have effects other than vasodilatation, these effects are an intrinsic part of this perfusion model and add to its qualities and deficits as a whole. We cannot distinctly determine, whether the dual vessel approach or the vasoldilatation by metamizole has greater beneficial impact. However, we show that a perfusion model that combines both accomplishes successful perfusion and then seems superior to the common single vessel model. 

We revised the “Discussion” part of our manuscript to make this clearer. (“Discussion” part, p. 22: “We show, that dNEVLP with sufficient vasodilatation by metamizole application on demand seems to be superior to sNEVLP in terms of organ preservation.”)

I our opinion, the above mentioned facts and considerations make the addition of a sNEVLP + metamizole group unnecessary. In compliance with the 3R-principles for responsible animal experiments we therefore decided, not to include this group in our work.

“In addition, oxygen consumption and oxygen saturation levels at inlet and effluent would be important to evaluate, since providing oxygen via both vessels is a key benefit of dual perfusion based on human liver perfusion literature.”

Thank you for your remark. Oxygen saturation was measured in our experiments. We calculated the oxygen consumption and included the results in our manuscript 

“Methods” part, p. 11: “Oxygen uptake was calculated according to Tolboom et al.”

“Results” part, p.14: “Oxygen consumption remained high throughout perfusion in all groups without significant differences (dNEVLPMP: VO2T0 = 0.03 ml/min/g, VO2T3 = 0.04 ml/min/g, VO2T6 = 0.03 ml/min/g; dNEVLPMH: VO2T0 = 0.03 ml/min/g, VO2T3 = 0.04 ml/min/g, VO2T6 = 0.05 ml/min/g; dNEVLP-M: VO2T0 = 0.04 ml/min/g, VO2T3 = 0.05 ml/min/g, VO2T6 = 0.04 ml/min/g; sNEVLP: VO2T0 = 0.02 ml/min/g, VO2T3 = 0.04 ml/min/g, VO2T6 = 0.03 ml/min/g; VO2T0 p = 0.68, VO2T3 p = 0.19, VO2T6 = 0.91).“

„Discussion“ part, p. 20: „Although oxygen uptake did not show a significant difference...“).

Our data show that oxygen consumption was high throughout perfusion in all four groups without significant differences. Overall, the dual vessel groups show a trend towards higher oxygen consumption, however, no statistical significance could be shown. As mentioned in the manuscript, only 9-12% of rat liver parenchyma are solely supplied by the hepatic artery. Therefore, differences in oxygen consumption between sNEVLP and dNEVLP can be expected to be very small and difficult to detect. Since oxygen saturation in the arterial and portovenous flow area is the same in our perfusion model, we do not expect higher oxygen supply to be the key beneficial factor of dNEVLP. We much more think that it`s positive effects rely on the perfusion of areas that are not reached by the portal vein, such as the bile duct. These areas may be small (and might therefore not significantly contribute to the livers total oxygen consumption) but are of great importance for the organs viability. Our results show that dNEVLP does make a difference as both liver and bile duct tissue were significantly better preserved.

“Another suggestion is to use some of the clinical perfusion viability criteria already used, such as the one published by Mergental et al to compare the viability of the perfused grafts.”

Mergental et al. (7) proposed two major and three minor criteria for the viability of human livers after two hours of NMP: Major viability criteria were perfusate lactate levels lower than 2.5 mmol/L (= 22.52 mg/dL) and the presence of bile production. Minor viability criteria were pH of greater than 7.3, homogenous perfusion with soft consistency of the parenchyma and stable arterial and portovenous flows of 150 ml/min and 500 ml/min, respectively. For the human liver with a total weight of around 1500 g these flow rates result in 0,1 ml/min/g arterial and 0.33 ml/min/g portovenous flow. Mergental et al. suggested that perfused livers should be judged viable if they met one or more major and two or more minor criteria.

Eventhough lactate clearance is difficult to assess due to our dialysis circuit, our (supplementary) data show that even after six hours of normothermic perfusion, all perfused livers met at least one major criterion (bile production) and 2 minor criteria (stable flows and homogenous perfusion). Therefore, all livers in our experiments had to be estimated as viable and transplantable. We added this to our discussion. However, a comparison between the human system used by Mergental et al. and ours is difficult, especially due to the fact that using criteria developed for human livers to assess rat livers is questionable and implicates numerous limitations. (“Discussion” part: p. 22: “Moreover, all livers in our experiments met the clinical criteria for viability and transplantability as established by Mergental et al....”)

“A more technical concern that could potentially influence the perfusion metrics and overall outcome was ligating the SHVC and cannulating the IVC. This can restrict the outflow of the perfusate and can increase the overall pressure within the liver. Research has shown that even partial occlusion of veins in the liver can lead to diffused hepatic congestion and enlargement (ie. sinusoidal thrombosis).”

Ligating the SHVC and cannulating the IHVC is an essential step in creating a closed perfusion circuit. This technique is also used in human normothermic machine perfusion of the liver (e.g. described by Ravikumar et al. using the Organox Metra Device)(8). Moreover, the diameter of the cannula that we used for the IHVC outflow was large enough not to downsize the lumen of the vessel. In our experiments, histopathology showed no signs of sinusoidal congestions, thrombosis or any other signs of pressure damage other than those caused by excessive arterial pressures in the dNEVLP-M group. Also, we never witnessed any problems using this technique in our previous publications (5, 9).

“The final technical concern is the flow rates that were used in this study. For dual vessel perfusion the flow was set to 1.1mL/min/g (paragraph 201-202) generating a flow of 1ml/min/g liver through the portal and 0.1 ml/min/g liver through the hepatic artery. These flow rates are very low, compared with other papers in literature which listed the arterial flow from 0.21 +/- 0.02 to 3.5 +/- 0.2 mL/min/g liver, and portal flow 1.53 +/- 0.19 to 32.1 +/- 1.6 mL/min/g liver. The flow rates stated in the manuscript are consistent with mouse liver perfusions. With such low flow rates, the concern is about the availability of oxygenated perfusate to the liver and if the flow rates were high enough to fully oxygenate the organ. Without data on the oxygen consumption and saturation values, it is not possible to ascertain if the organ was oxygenated sufficiently.”

Thank you for your remark on this important aspect. The flow rates were chosen based on our previous experiences with rat liver perfusion (5, 9) In our previous works we have shown that a portovenous flow rate of 1 ml/min/g liver accomplishes sufficient liver perfusion. Establishing our dual vessel NEVLP model, we found that the arterial flow rate of 0.1 ml/min/g liver achieved sufficient arterial perfusion without causing notable damage. Since there is no literature on dual vessel NEVLP of rat livers for more than four hours, we chose the flow rate that worked best for six hours of dNEVLP in our preliminaries. Other publications on dual vessel NEVLP either show arterial pressures but lack data on flow rates (10) or vise versa (11.). Our data on oxygen consumption show that sufficient oxygenation was accomplished for all organs. These results support our results in bile production, urea production and histopathology that show sufficient organ perfusion.

“1 mL Lactated Ringers that is supplemented with 500 IE Heparin is injected into the IVC. This concentration of Heparin in rats weighing between 280-350g will subsequently euthanize the rat exposing the liver to warm ischemic time (WIT). The portal and aorta are then cannulated and flushed after with 20 mL of 4C HTK, however the liver is still within the body cavity, and in the subjected to room temperature while the hepatic artery SHVC, right suprarenal vein, esophageal veins are ligated and the IVC is cannulated. Even though the WIT time is likely minimal, it should be clarified in the manuscript.”

Thank you for your remark. The rat was euthanized by blood withdrawal immediately after the injection of Lactated Ringer´s and heparine. We clarified the warm ischemia time (WIT) in our manuscript. (“Methods” part, p.10: “Time between blood collection and final mobilization of the liver (warm ischemia time, WIT) did not exceed 15 minutes.”)

“Average initial pressure was 5.65 mmHg. This pressure seems awfully high given that the temperature of the liver is close to 4C at the time of connection? This could cause damage to the endothelial layer within the vasculature of the liver and subsequently cause further complications during the length of perfusion.”

Thank you for giving us the chance to clarify this point. After the liver was connected to the perfusion circuit flows were slowly increased over a rewarming phase of 15 minutes. T0 was set when full flow was reached and “initial pressures” were then measured. We altered the manuscript to clarify this (“Methods” part, p. 11: “After connection to the perfusion circuit flows were slowly increased over a rewarming period of 15 minutes. T0 was set when full flows were reached.”)

“Supplemental data shows some lactate clearance, but it is unclear if the dialysis cartridge used removes lactate from the perfusate.”

Thank you for your remark. Average lactate levels in the dialysate rose from 2,3 mg/dL to 5,4 mg/dL without significant differences between the groups. We therefore attribute differences in perfusate lactate levels to differences in lactate clearence. However, since this is difficult to assess reliably, no conclusions regarding lactate clearance were drawn.

---

## [Decision Letter · Decision Letter 1]

8 Jun 2020

PONE-D-20-06139R1

Dual versus single vessel normorthermic ex vivo perfusion of rat liver grafts using metamizole for vasodilatation.

PLOS ONE

Dear Dr. Sauer,

Thank you for submitting your manuscript to PLOS ONE. After careful consideration, we feel that it has merit but does not fully meet PLOS ONE’s publication criteria as it currently stands. Therefore, we invite you to submit a revised version of the manuscript that addresses the points raised by reviewer 3, in particular please comment on edema.

We look forward to receiving your revised manuscript.

Kind regards,

Michael Bader

Academic Editor

PLOS ONE

Reviewers' comments:

Reviewer's Responses to Questions

**Comments to the Author**

1. If the authors have adequately addressed your comments raised in a previous round of review and you feel that this manuscript is now acceptable for publication, you may indicate that here to bypass the “Comments to the Author” section, enter your conflict of interest statement in the “Confidential to Editor” section, and submit your "Accept" recommendation.

Reviewer #1: All comments have been addressed

Reviewer #2: All comments have been addressed

Reviewer #3: (No Response)

2. Is the manuscript technically sound, and do the data support the conclusions?

Reviewer #1: Yes

Reviewer #2: Yes

Reviewer #3: Yes

3. Has the statistical analysis been performed appropriately and rigorously? 

Reviewer #1: I Don't Know

Reviewer #2: Yes

Reviewer #3: Yes

4. Have the authors made all data underlying the findings in their manuscript fully available?

Reviewer #1: Yes

Reviewer #2: Yes

Reviewer #3: Yes

5. Is the manuscript presented in an intelligible fashion and written in standard English?

Reviewer #1: Yes

Reviewer #2: Yes

Reviewer #3: Yes

6. Review Comments to the Author

Reviewer #1: I was satisfied with the answers given by the authors that meet my questions. The placement of the figures values the work. Congratulations.

Reviewer #2: (No Response)

Reviewer #3: The authors did address some key comments that were raised in the original manuscript, and the detail provided in the responses is appreciated. I also appreciated the nuances observed in human vs rat comparisons.

I have two minor comments: First, the question re. ATP from two reviewers was not addressed. I do appreciate that performing such analyses may require repeat of most if not all experiments which in my opinion is not necessary and would likely be unnecessary duplication that would not justify the use of research animals; one reason the Mergenthal criteria was recommended as an alternative to ATP is it can be evaluated practically. Since the authors have already done this in the revision, I would have recommended the authors to note this in their rebuttal so it does not appear like an ignored comment to the reviewers.

My second comment would be if the authors could shed some light on if they noticed any difference in relation to edema (or lack thereof) between the groups? Noting that pressures in dNEVLP in the metamizole on-demand group were better, I would be interested to see if this correlates in the context of edema as well.

Overall, I consider the manuscript is acceptable for publication with very minor revisions.

7. PLOS authors have the option to publish the peer review history of their article (what does this mean?). If published, this will include your full peer review and any attached files.

Reviewer #1: Yes: Rui Miguel Martins

Reviewer #2: No

Reviewer #3: No

---

## [Author Response · Author response to Decision Letter 1]

17 Jun 2020

Dear Dr. Heber,

Thank you very much for giving us the opportunity to submit a newly revised version of our manuscript. We hereby submit our revised manuscript as clean copy (Manuscript.docx) and additional copy with the changes being done highlighted (Revised Manuscript with Track Changes.docx). Please find below our specific answers to the remaining questions and suggestions raised by Reviewer #3 as a point-to-point response. We hope that our manuscript is appropriate for publication in PLOS ONE.

Yours sincerely,

Igor M. Sauer.

Reviewer #3

I have two minor comments: First, the question re. ATP from two reviewers was not addressed. I do appreciate that performing such analyses may require repeat of most if not all experiments which in my opinion is not necessary and would likely be unnecessary duplication that would not justify the use of research animals; one reason the Mergenthal criteria was recommended as an alternative to ATP is it can be evaluated practically. Since the authors have already done this in the revision, I would have recommended the authors to note this in their rebuttal so it does not appear like an ignored comment to the reviewers.

Thank you for pointing this out. We apologize for the negligence. 

My second comment would be if the authors could shed some light on if they noticed any difference in relation to edema (or lack thereof) between the groups? Noting that pressures in dNEVLP in the metamizole on-demand group were better, I would be interested to see if this correlates in the context of edema as well.

Thank you for your remark and we apologize for not addressing this aspect in our first rebuttal. The assessment of edema in our work was limited to the histopathological examination of the livers. The pathologist did not find relevant edema in any of the four groups. We now mention this in the manuscript (“Results” part, p. 13: “Histopathology did not show relevant edema in any of the four groups after perfusion.” and “Discussion” part, p. 19: “Interestingly, the high arterial pressures did not lead to relevant development of edema.”)

The assessment of edema by lyophilisation is currently being established in our workgroup. Preliminary results correlate with the results from our histopathological examination and do not show significant edema after neither single vessel nor dual vessel perfusion when compared to native livers. These results indicate that the development of edema is not a relevant problem in our small animal NEVLP model. However, we will further investigate this aspect in our future projects and thank you for your constructive feedback.

---

## [Editor Report · Decision Letter 2]

19 Jun 2020

Dual versus single vessel normothermic ex vivo perfusion of rat liver grafts using metamizole for vasodilatation.

PONE-D-20-06139R2

Dear Dr. Sauer,

We’re pleased to inform you that your manuscript has been judged scientifically suitable for publication and will be formally accepted for publication once it meets all outstanding technical requirements.

Kind regards,

Michael Bader

Academic Editor

PLOS ONE
---

## [Editor Report · Acceptance letter]

24 Jun 2020

PONE-D-20-06139R2 

Dual versus single vessel normothermic ex vivo perfusion of rat liver grafts using metamizole for vasodilatation. 

Dear Dr. Sauer:

I'm pleased to inform you that your manuscript has been deemed suitable for publication in PLOS ONE. Congratulations! Your manuscript is now with our production department. 

Kind regards, 

on behalf of

Prof. Michael Bader 

Academic Editor

PLOS ONE